## Perspective

human-computer interaction

fake science, trust in science, knowledge crisis

**Author for correspondence:**
Stephen A. Matlin
e-mail: s.matlin@imperial.ac.uk

This article has been edited by the Royal Society of Chemistry, including the commissioning, peer review process and editorial aspects up to the point of acceptance.

# Fake science and the knowledge crisis: ignorance can be fatal

Henning Hopf[1,2], Alain Krief[1,3,4], Goverdhan Mehta[1,5] and Stephen A. Matlin[1,6]

[1]International Organization for Chemical Sciences in Development, 61 rue de Bruxelles, 5000 Namur, Belgium
[2]Institute of Organic Chemistry, Technische Universität Braunschweig, Braunschweig 38106, Germany
[3]Chemistry Department, Namur University, 5000 Namur, Belgium
[4]HEJ Research Institute, University of Karachi, Karachi City, Sindh 75270, Pakistan
[5]School of Chemistry, University of Hyderabad, Hyderabad 500046, India
[6]Institute of Global Health Innovation, Imperial College London, London SW7 2AZ, UK

HH, 0000-0001-7040-6506; AK, 0000-0002-9223-1644;
GM, 0000-0001-6841-4267; SAM, 0000-0002-8001-1425

Computers, the Internet and social media enable every individual to be a publisher, communicating true or false information instantly and globally. In the 'post-truth' era, deception is commonplace at all levels of contemporary life. Fakery affects science and social information and the two have become highly interactive globally, undermining trust in science and the capacity of individuals and society to make evidence-informed choices, including on life-or-death issues. Ironically, drivers of fake science are embedded in the current science publishing system intended to disseminate evidenced knowledge, in which the intersection of science advancement and reputational and financial rewards for scientists and publishers incentivize gaming and, in the extreme, creation and promotion of falsified results. In the battle for truth, individual scientists, professional associations, academic institutions and funding bodies must act to put their own house in order by promoting ethics and integrity and de-incentivizing the production and publishing of false data and results. They must speak out against false information and fake science in circulation and forcefully contradict public figures who promote it. They must contribute to research that helps understand and counter false information, to education that builds knowledge and skills in assessing information and to strengthening science literacy in society.

# 1. Introduction

Seek truth from facts [1]
实事求是; shí shì qiú shì
Quoted from the Han Dynasty *Book of Han*, AD 111

Ignorance of the truth, or knowledge that is not acted upon, can be fatal. This basic principle applies at levels from personal to planetary. Fakery affects science as well as everyday social information and, since the two have become highly interactive globally, a vicious cycle is now operating on an increasing scale. The fake news/fake science cycle undermines the credibility of science and the capacity of individuals and society to make evidence-informed choices in their best interests.

At the individual level, lack of reliable knowledge about how to maintain personal physical, nutritional and health security can result in avoidable harm or death. An illustration is the child illnesses, permanent disabilities and deaths that have resulted worldwide from the fabricated scientific report [2,3] that the measles, mumps and rubella (MMR) vaccine causes autism. Despite the proven record of efficacy of vaccines in preventing infections by deadly diseases, the widespread dissemination of this lie, especially through social media, resulted not only in record levels of measles infections in Europe [4] in 2018 but has added fuel to a growing, broader phenomenon known as 'vaccine hesitancy' [5,6].

At a collective level, false information can alter attitudes and policies on crucial ecological, social and political issues and, in the extreme, can place entire populations at national, regional and even global levels at risk of harm. For example, the denial of anthropogenic climate change, dismissed without counter-evidence as 'fake science', has resulted in the international agreement on climate change losing universal acceptance and its impact on the level of global warming is likely to have disastrous consequences worldwide in the twenty-first century [7]. Portrayal of ethnic groups, foreigners or foreign states as enemies through fake stories is an old technique which has been given new potency by modern methods of mass communication and can provoke genocides and wars [8]. Fake Twitter accounts have reportedly been used to send millions of messages aiming to influence attitudes towards Brexit in the 2016 UK referendum and views on candidates in the 2016 US presidential election [9].

## 2. Validated information needs to be acted upon

Awareness of the evidence that smoking causes severe illnesses and that tobacco kills up to half of its users has not yet enabled the 1.1 billion smokers around the world to quit their addiction or governments to introduce outright bans on smoking. Denial, fake news, the deliberate undermining of true data by portrayal as 'junk science' to distort public health policy, fabricated information, distortion of media framing and covert illicit trading have been documented in the decades-long battle by the tobacco industry and its supporters to sustain their lucrative but fatal trade [10,11].

Examples such as climate change and tobacco illustrate the difficulties that both individuals and society can have in determining what is factually correct, how to recognize the biases and vested interests that may lie behind information available and how to balance risks against benefits at personal, national and global levels.

## 3. Evolution of scientific validation

A critical factor is the question of who is accorded authority to determine reliability of facts and make judgement about the veracity of information on offer. Since ancient times, those with wealth, power and elevated hierarchical positions were treated as privileged sources, as were some individuals regarded as selfless seekers of wisdom in the domains of spirituality, scholarship or science.

Since its introduction by Francis Bacon (1561–1626), the evolved scientific method has included unbiased observations that are evaluated for reproducibility and subjected to careful self-criticism and logical thought about their meaning and implications, then offered for inspection by the world at large. The establishment of learned societies and the publication of journals, beginning with the *Philosophical Transactions of the Royal Society*, first published in 1665, provided a mechanism for presenting information that could be critically examined by the scientific community. If disproved, the prevailing models and theories would be replaced by new ones that were more consistent with the contemporary state of knowledge. This provisional character of science is not a weakness but is one of the key reasons for its strength.

The evolution of this process, into the second half of the twentieth century, established a 'gold standard' for the reliability of knowledge. It has been the bedrock of the esteem in which science has been held, as an honest and impartial source of evidence-based knowledge, not only to advance the frontiers of the field but also to inform the public and politicians and aid their decision-making. The

1918 landmark report by Richard Burdon Haldane to the British Prime Minister signalled the strength of the evolving relationship between science and policy, with Haldane arguing for the principle that politicians should stay out of decisions about research funding, listen to expertise, take time to think and reflect before reaching a conclusion and, when asking scientists for advice, resist telling them what that advice should be [12].

# 4. The changing landscape: the revolution in knowledge production

While the degree to which scientific inputs to policy formulation are important continues to be debated, the current revolution in knowledge production has further complicated the issue. An illustration of the extent to which the landscape has changed was seen in the centenary year of the 'Haldane Principle', which witnessed a major scientific report commissioned by the US government and issued by 13 federal agencies—warning of the consequences of climate change and therefore at odds with the Administration's policies—being rejected by a number of leading politicians, including the President, on the grounds that they 'don't believe it', while they (baselessly) accused climate scientists of being driven by money [13,14].

In 1991, Harnad described four stages in the means of production of knowledge in human beings [15]. The first three were the emergence of language (hundreds of thousands of years ago) and invention of writing (several thousand years ago) and printing (over 500 years ago). The fourth had only very recently begun, with the invention of the Internet and the capacity it provides for anyone in the world to be a publisher—to communicate any information they wish, true or false, instantly and globally.

Facts and their denial are no longer determined by any type of authority, but in principle by every individual, regardless of his or her education and reputation or studiously acquired knowledge of a field. The manipulation of data by anyone (including scientists) becomes ever easier. Due to the ready availability of information and communication technology (ICT) tools and access to the Internet and social media, there now exist countless ways to create and distribute products of unknown veracity, including manipulated textual and pictorial material. The predictions of the anarchistic philosopher of science Paul Feyerabend that 'anything goes' and of the conceptual artist Joseph Beuys that 'every human being is an artist' have thus become reality.

In his 1943 essay on the Spanish Civil War, the writer George Orwell recognized the way that people in politics and wars make use of the available propaganda mechanisms to create their own versions of the truth, expressing his fear that 'the very concept of objective truth is fading out of the world' [16]. This ongoing challenge has been exacerbated and accelerated greatly by ICT and the fourth revolution in knowledge production. As Harnad recognized, each of these knowledge revolutions represented a profound, qualitative change both in HOW human beings communicate and think and in WHAT is thought.

# 5. Consequences for science and for science publishing and assessment

The impacts of the fourth revolution, barely to be seen three decades ago, are now dramatically evident, including in contemporary language. A signpost was the declaration by Ralph Keyes in 2004 that 'we live in a post-truth era'—a stage of social evolution that is 'beyond honesty', in which 'deception has become commonplace at all levels of contemporary life' [17]. More recent signposts have been the emergence in 2017 of the term 'alternative facts' to describe inaccurate data and the designation of 'truth isn't truth' as the 2018 Quote of the Year.

There are growing impacts both on the interface between science and society and within the domain of science itself. It has been argued that, in the current political and media environment, 'distrust in the scientific enterprise and misperceptions of scientific knowledge increasingly stem... from the widespread dissemination of misleading and biased information' [18]. The philosopher Bruno Latour has observed that 'facts remain robust only when they are supported by a common culture, by institutions that can be trusted, by a more or less decent public life, by more or less reliable media' [19]. While surveys about the public's view of the trustworthiness of scientists produce results that vary with time and place [20,21], in his 2017 book on the 'death of expertise' Tom Nichols described the many forces trying to undermine the authority of 'experts' [22], so that the term itself has started to be used in a contemptuous way to justify dismissing their advice [23].

Facing this challenge, it is especially important that the scientific world as a whole upholds the highest standards of ethical behaviour, honesty and transparency, aiming to sustain the gold standards of research integrity and validated information. Sadly, a range of forces are working counter to this aspiration. People in the world of science are not immune from the personal ambitions and prevailing pressures that drive behaviour in general.

As recently described [24], three closely inter-related sub-systems (science advancement, reputational rewards and financial returns) collectively form an overall scientific publishing system that has become heavily flawed. It encourages scientists to distort and exaggerate their results in striving for new grants, promotions and distinctions; and encourages publishers to cherry-pick work, hype results and distort refereeing in the competition for high status and correspondingly high profits from publication charges. Both authors and publishers are incentivized to game the system to their mutual advantage. In the extreme, the perverse incentives generated result in authors fabricating data, predatory journals hunting for papers and fake journals being created that seek only the authors' fees for article processing.

The scale of the fake science problem is becoming increasingly evident. The percentage of scientific articles retracted because of fraud has increased by an order of magnitude since 2000 and high rates of retraction are seen for the most prestigious journals, illustrating both the extent to which flawed claims are perpetrated by scientists seeking prominence and weaknesses and even fakery in the current practice of peer reviewing [25]. A recent investigation of publishing in predatory 'open access' journals and fake conferences has revealed a global ecosystem of predatory publishers churning out 'fake science' for profit [26]. The intrusion of such journals into the traditionally respected space of science publishing seriously undermines the integrity and credibility of science and, if not stopped and sanctioned immediately, could turn out to be fatal for the field as we know it.

It is a fundamental strength of the scientific system that knowledge that is incorrect will eventually be discovered and discarded. However, the pace and scale at which material that is at best dubious and at worst deliberately false is now being published is creating a crisis. The consequences are very damaging for the science enterprise, with a loss of respect for the results of science and the scientific method leading, inter alia, to a steep decline in funding, jobs and students wishing to enter the field. The crisis is also damaging society, creating an 'anything goes' environment in which 'alternative facts' are not tested and decisions affecting the lives of people everywhere are not informed by authentic data or valid conclusions. Thus, in the new age of the fourth revolution in the means of production of knowledge, scientific publishing has become a part of the problem of fake news, rather than a bulwark against it.

# 6. Ways forward

Fake science and fake news are complex phenomena involving a variety of causes, channels of dissemination and consequences. Solving the challenges they pose will not be accomplished by a single approach or simple set of measures, but will require concerted effort by a wide range of actors across sectors.

To address the general societal problem of fake news, several initiatives now underway or being discussed offer promising approaches. Apart from those directly involving science and scientists, which are discussed separately below, they include the following.

Efforts are needed to counter the spread of false information via social media, through modifications to computer algorithms that favour 'trending' of stories without a factual basis [27], and development of tools that help identify and build skills in recognizing false claims [28–30]. The limitations of large-scale automated approaches and the ingenuity with which they can be gamed must, however, be recognized [31].

There should be more efforts to increase the responsibility taken by social media services for the content they permit online. The fundamental issue of whether social media should be regarded as 'platforms' that are not responsible for content (as the social media maintain) or as 'publishers' who can, like traditional print publishers, be held liable for the content they disseminate (as some critics of the present position propose), with many legal, regulatory, financial, ethical and operational ramifications, remains in dispute [32,33]. Meanwhile, there has been widespread dissatisfaction with the results of self-regulation by social media to date, and highly publicized failures in areas including politics, racism and health have led to calls for more regulation and/or more action by social media [34,35]. Initiatives required include efforts to increase the speed and scope of measures to remove offensive and injurious materials and to develop algorithms to detect and exclude fraudulent sources.

Scientists must not remain bystanders in the battle against fakery in news generally as well as in their own domains of expertise. They can contribute to understanding the phenomenon of fake news, which

has typically been studied along four lines: characterization, creation, circulation and countering [36]. Multidisciplinary effort is needed to understand better how the Internet spreads content and how readers process the news and information they consume, as well as how social media platforms are manipulated to amplify particular stories through the use of fake accounts and 'bots' [37–40]. As an example, WhatsApp has selected 20 research teams worldwide, including from India, to work towards understanding how misinformation spreads and what additional steps the mobile messaging platform could take to curb fake news [41].

Scientists must be willing to speak out when they see false information being presented in social media, traditional print or broadcast press [42]. They must use these media fully themselves [43] to offer facts and evidence in succinct layman's language while emphasizing the breadth and depth of the scientific consensus which underpins the present state of knowledge and pointing to the lack of scientific rigour in the false information [44–46]. They must be willing to contradict public leaders and opinion formers who condemn or dismiss valid science without offering verified evidence of their own, as has happened, for example, in the USA and India [47–49].

For the longer term, scientists must be better advocates for and contributors to the generation of a more scientifically literate society [50]. The ultimate defence against fake facts is the capacity of each individual to examine critically the information on offer and to reach judgement about its trustworthiness that is based on evidence and reasoning. Scientists can contribute to inculcating 'scientific temper' in society. This term, coined in 1946 by Jawaharlal Nehru, describes a way of life, a process of thinking and acting which uses the scientific method and may, consequently, include questioning, observing, testing, hypothesizing, analysing and communicating [51].

The role of journalism remains important and development by scientists of stronger links with reputable journalists can encourage clearer and more accurate reporting of research [52].

Within the domain of science itself, individually and collectively through their professional associations, academic institutions and funding bodies, scientists must act in order to put their own house in order, through promoting ethical practices and research integrity, dealing with the problems of reproducibility and retractions [53], developing policies and practices to de-incentivize the production and publishing of false data and results and the use of 'predatory' journals that have inadequate peer review, and making maximum use of emerging artificial intelligence capacities [54] to detect and expose falsified data and images. Examples where measures are already being adopted or explored include India's use of a 'white list' to discourage researchers from publishing in predatory journals [55].

Education—both broadly as part of the development of life skills and specifically in the culture and methods of science—is an essential part of the long-term solution, so that young people are equipped with knowledge, skills and tools to be able to critically examine information and assess its veracity [56,57]. As noted by the President of the European Research Council, 'We need to train a new generation of critical minds. Science is not about learning facts by heart, established long ago; it is about knowing how to call into question and move forward. The majority of youth rely mostly on social media to get their news, so we must tackle this issue through improved news literacy, and it is the task of our educators and society at large to teach children how to use doubt intelligently and to understand that uncertainty can be quantified and measured' [58].

Research indicates that pre-emptively inoculating people before they receive misinformation (prebunking) is more effective than refutation after receipt (debunking) in reducing the influence of misinformation. Synthesizing separate lines of research from education, cognitive psychology and inoculation theory (a branch of psychological research) provides a coherent set of recommendations for educators and communicators. Scientific explanations that involve clear communication of scientific concepts and the current scientific consensus are ideally coupled with inoculating explanations of how that science can be distorted [59].

Data accessibility. All data cited in this article are taken from the sources given in the references.

Authors' contributions. All authors listed made substantial contributions to the conception, design, drafting and revision of this article; gave final approval of the version to be published; and agreed to be accountable for all aspects of the work in ensuring that questions related to the accuracy or integrity of any part of the work are appropriately investigated and resolved.

Competing interests. We declare we have no competing interests.

Funding. We are grateful to the International Organization for Chemical Sciences in Development, the Gesellschaft Deutscher Chemiker, the Royal Society of Chemistry and Syngenta for supporting a workshop held in the Indian Institute of Chemical Technology, Hyderabad and hosted by its Director, Dr Srivari Chandrasekhar in January 2019, during which this article was prepared.

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
