## [Reviewer comments · Royal Society Open Science]

Review History

RSOS-190161.R0 (Original submission)

Review form: Reviewer 1 (David Spiegelhalter)

Is the manuscript scientifically sound in its present form?

No

Are the interpretations and conclusions justified by the results?

No

Is the language acceptable?

Yes

Is it clear how to access all supporting data?

Not Applicable

Do you have any ethical concerns with this paper?

No

Have you any concerns about statistical analyses in this paper?

No

Recommendation?

Reject

Comments to the Author(s)

This is a polemical piece, with somewhat exaggerated and non-scientific arguments about a 'post-truth' society - eg trust in scientists remains extremely high. But I don't mind about that as it is clearly an opinion piece, and I agree with its opinion that science should put its own house in order, particularly by recognising that distortions and exaggerations occur in the literature not through the obvious financial conflicts, but through scientists advancing their reputations and personal agendas.

As an polemical opinion piece, I like the paper, but I have no idea if this is what the journal wants.

Review form: Reviewer 2

Is the manuscript scientifically sound in its present form?

Yes

Are the interpretations and conclusions justified by the results?

Yes

Is the language acceptable?

Yes

Is it clear how to access all supporting data?

Yes

Do you have any ethical concerns with this paper?

No

Have you any concerns about statistical analyses in this paper?

No

Recommendation?

Accept with minor revision (please list in comments)

Comments to the Author(s)

This paper deals with a critical issue for society, as well as for society. However, as currently written it ends much too weakly, in two respects:

1). Although elsewhere the critical importance of new ways of communicating through social media is emphasized, this point is completely ignored in the recommendations for scientists to "speak out". The authors may not be active on social media themselves, being of an older generation. If so, I suggest that they consult with their younger colleagues in order to add specific suggestions for how scientists should get engaged with new communications tools, in order not to become increasingly invisible to most of society.

2). More surprisingly, the advice for scientists in the last "A Way Forward" is much too brief, lacking the type of specificity one would expect from the authors, each of whom should have specific ideas based on their nation's experiences. For example, how are the leading scientists in India trying to confront that nation's serious problems with new journals? Can such information and experiences be generalized to make suggestions for other nations?

Decision letter (RSOS-190161.R0)

01-Mar-2019

Dear Dr Matlin:

Title: Fake science and the knowledge crisis: Ignorance can be fatal
Manuscript ID: RSOS-190161

Thank you for submitting the above manuscript to Royal Society Open Science. On behalf of the Editors and the Royal Society of Chemistry, I am pleased to inform you that your manuscript will be accepted for publication in Royal Society Open Science subject to minor revision in accordance with the referee suggestions. Please find the reviewers' comments at the end of this email.

The reviewers and handling editors have recommended publication, but also suggest some minor revisions to your manuscript. We have considered the points raised by Reviewer 1 and are satisfied that this type of perspective is suitable for the journal. Therefore, I invite you to respond to the comments (in particular, those of Reviewer 2) and revise your manuscript.

Because the schedule for publication is very tight, it is a condition of publication that you submit the revised version of your manuscript before 10-Mar-2019. Please note that the revision deadline will expire at 00.00am on this date. If you do not think you will be able to meet this date please let me know immediately.

- 1) A text file of the manuscript (tex, txt, rtf, docx or doc), references, tables (including captions) and figure captions. Do not upload a PDF as your "Main Document".
- 2) A separate electronic file of each figure (EPS or print-quality PDF preferred (either format should be produced directly from original creation package), or original software format)
- 3) Included a 100 word media summary of your paper when requested at submission. Please ensure you have entered correct contact details (email, institution and telephone) in your user account

- 4) Included the raw data to support the claims made in your paper. You can either include your data as electronic supplementary material or upload to a repository and include the relevant doi within your manuscript
- 5) All supplementary materials accompanying an accepted article will be treated as in their final form. Note that the Royal Society will neither edit nor typeset supplementary material and it will be hosted as provided. Please ensure that the supplementary material includes the paper details where possible (authors, article title, journal name).

Best wishes,
 Dr Laura Smith
 Publishing Editor, Journals

RSC Associate Editor:
 Comments to the Author:
 (There are no comments.)

RSC Subject Editor:
 Comments to the Author:
 (There are no comments.)

Reviewer comments to Author:
 Reviewer: 1

Comments to the Author(s)
 This is a polemical piece, with somewhat exaggerated and non-scientific arguments about a 'post-truth' society - eg trust in scientists remains extremely high. But I don't mind about that as it is clearly an opinion piece, and I agree with its opinion that science should put its own house in order, particularly by recognising that distortions and exaggerations occur in the literature not

through the obvious financial conflicts, but through scientists advancing their reputations and personal agendas.

As an polemical opinion piece, I like the paper, but I have no idea if this is what the journal wants.

Reviewer: 2

Comments to the Author(s)

This paper deals with a critical issue for society, as well as for society. However, as currently written it ends much too weakly, in two respects:

1). Although elsewhere the critical importance of new ways of communicating through social media is emphasized, this point is completely ignored in the recommendations for scientists to "speak out". The authors may not be active on social media themselves, being of an older generation. If so, I suggest that they consult with their younger colleagues in order to add specific suggestions for how scientists should get engaged with new communications tools, in order not to become increasingly invisible to most of society.

2). More surprisingly, the advice for scientists in the last "A Way Forward" is much too brief, lacking the type of specificity one would expect from the authors, each of whom should have specific ideas based on their nation's experiences. For example, how are the leading scientists in India trying to confront that nation's serious problems with new journals? Can such information and experiences be generalized to make suggestions for other nations?

Author's Response to Decision Letter for (RSOS-190161.R0)

See Appendix A.

Decision letter (RSOS-190161.R1)

19-Mar-2019

Dear Dr Matlin:

Title: Fake science and the knowledge crisis: Ignorance can be fatal
Manuscript ID: RSOS-190161.R1

It is a pleasure to accept your manuscript in its current form for publication in Royal Society Open Science. The chemistry content of Royal Society Open Science is published in collaboration with the Royal Society of Chemistry.

RSC Associate Editor
Comments to the Author:
(There are no comments.)

Reviewer(s)' Comments to Author:

Appendix A

ResponseToReviewers2019a

We are delighted that the reviewers and handling editors have recommended publication and greatly appreciate the reviewers' comments.

We thank Reviewer 2 for specific suggestions of ways to strengthen the paper, in particular by adding specific ideas for how scientists should get engaged with new communications tools, and by expanding the "Way Forward" section at the end and giving examples of how scientists, including in India, are confronting problems.

Response:

The final section, re-titled "Ways Forward" (title highlighted in yellow) has been completely re-written and substantially expanded, with a detailed discussion of how scientists can help to address both fake news in society and fake science, including through engaging with new communications tools. Examples of initiatives from a number of countries, including India, are cited in a significantly expanded reference list.

An additional sentence has been added to the Summary to reflect these changes and small modifications in other sections, including some additional insertions of recent literature, are highlighted in yellow in the revised text and reference list. In particular (p4) a short addition addresses the variability of public attitudes to scientists and 'experts'.